# Diagnostic Agreement between Physicians and a Consultation–Liaison Psychiatry Team at a General Hospital: An Exploratory Study across 20 Years of Referrals

**DOI:** 10.3390/ijerph18020749

**Published:** 2021-01-17

**Authors:** Mattia Marchi, Federica Maria Magarini, Giorgio Mattei, Luca Pingani, Maria Moscara, Gian Maria Galeazzi, Silvia Ferrari

**Affiliations:** 1Department of Biomedical, Metabolic and Neural Sciences, University of Modena and Reggio Emilia, Via Giuseppe Campi, 287–41125 Modena, Italy; federica.maria@hotmail.com (F.M.M.); luca.pingani@unimore.it (L.P.); gianmaria.galeazzi@unimore.it (G.M.G.); silvia.ferrari@unimore.it (S.F.); 2Department of Economics & Marco Biagi Foundation, University of Modena and Reggio Emilia, Via J. Berengario, 51–41121 Modena, Italy; giorgiomatteimd@gmail.com; 3Department of Mental Health, Modena Health Local Agency, Viale L.A. Muratori, 201-41124 Modena, Italy; mascaramaria@gmail.com

**Keywords:** consultation–liaison psychiatry, inter-rater agreement, psychosomatic medicine, general hospital

## Abstract

Consultation–liaison psychiatry (CLP) manages psychiatric care for patients admitted to a general hospital (GH) for somatic reasons. We evaluated patterns in psychiatric morbidity, reasons for referral and diagnostic concordance between referring doctors and CL psychiatrists. Referrals over the course of 20 years (2000–2019) made by the CLP Service at Modena GH (Italy) were retrospectively analyzed. Cohen’s kappa statistics were used to estimate the agreement between the diagnoses made by CL psychiatrist and the diagnoses considered by the referring doctors. The analyses covered 18,888 referrals. The most common referral reason was suspicion of depression (*n* = 4937; 32.3%), followed by agitation (*n* = 1534; 10.0%). Psychiatric diagnoses were established for 13,883 (73.8%) referrals. Fair agreement was found for depressive disorders (kappa = 0.281) and for delirium (kappa = 0.342), which increased for anxiety comorbid depression (kappa = 0.305) and hyperkinetic delirium (kappa = 0.504). Moderate agreement was found for alcohol or substance abuse (kappa = 0.574). Referring doctors correctly recognized psychiatric conditions due to their exogenous etiology or clear clinical signs; in addition, the presence of positive symptoms (such as panic or agitation) increased diagnostic concordance. Close daily collaboration between CL psychiatrists and GH doctors lead to improvements in the ability to properly detect comorbid psychiatric conditions.

## 1. Introduction

Consultation–liaison psychiatry (CLP) deals with the psychiatric care of ill or medically complex patients who have received admission to a general hospital (GH) for somatic reasons and who display psychiatric symptoms over the course of their hospital stay [1]. CLP practices and methodologies have developed in various ways throughout the second half of the 90s in different countries with different healthcare systems. Historically, consultation psychiatry (CP) has described requests for a second, expert opinion by a ward treating physician for a specific patient, while CLP has referred to a closer integration of psychiatry in a somatic department [2]. CLP has generally included regular psychiatric consultation hours, as well as routine guidance and supervision of the GH staff administering somatic treatment by a psychiatrist regarding psychiatric and psychosomatic themes [3,4,5,6,7]. CL psychiatrists operationalize the dialogue between psychiatry and the rest of medicine, connecting GH departments with community mental health services. Despite growing evidence suggesting that CLP services can be cost-effective because, among other things, they reduce the length of patients’ hospital stay [8,9], such an extensive framework may be demanding to apply [10]. In Italy, only a few CLP services have been formally and administratively recognized in GHs, as CP services are a more common framework [11,12].

Medical–psychiatric comorbidity has huge impacts on health, functioning, and quality of life in patients and caregivers, as well as on the organization of the health care system; nevertheless, psychiatric disorders continue to be underdiagnosed and misdiagnosed at GHs [1,13,14]. Studies of CLP have investigated this and found that it may be related to the culture and training of the multiprofessional GH staff, to pitfalls in the organization of health care systems and related funding strategies, or to the intrinsic clinical complexity of medical and psychiatric multi-morbidity [15,16]. Ward doctors in GHs are often overwhelmed by having to cope with both understaffing and high turnover, leading them to struggle to find the time and the ability to establish the intimate doctor–patient relationship that is essential for performing effective psychological assessments. Moreover, some doctors at GHs may feel uneasy in dealing with psychiatric conditions. In these cases, medical–psychiatric morbidity is likely to be neglected within the GH [13].

The involvement of clinical psychologists and specialists in psychosomatic medicine has led to closer examination of medical–psychiatric comorbidity within GHs, however, rates of referrals to psychiatry are generally low [17], and few studies have investigated and discussed the accuracy of initial diagnostic classification made by the GH doctors in patients subsequently evaluated by consulting psychiatrists. These few showed variable agreement and disagreement rates dependent on multiple factors, such as the diagnostic category [18,19], the age of the patient [20], and the professional experience of the physician [21]. Evaluating the diagnostic agreement between treating physicians and CL-psychiatrists with a dedicated CLP service may provide an index of the appropriateness of the referral, reflecting how medical–psychiatric comorbidity is acknowledged and managed within GH wards and providing valuable feedback to CL psychiatrists engaged in improving the dialogue between psychiatry and the other medical branches.

We evaluated patterns in psychiatry morbidity in the CLP service of a GH, assessing the reasons for referrals and the diagnostic concordance between referring GH doctors (whether physicians or surgeons) and the CL psychiatrists. We also investigated differences in diagnostic agreement, according to the features of the referral (i.e., whether the request is urgent or not urgent), of the patient (i.e., gender, and whether there is a positive or negative psychiatric history), of the referring ward (i.e., whether the request comes from internal medicine or a surgical ward), and of the CLP service (i.e., whether it is within its first or second 10 years of activity).

Taking into consideration the available literature, we hypothesized that affective disorders would be the most frequent reason for referral and that these referrals would have low diagnostic accuracy [18,19,20,22,23,24,25]. Moreover, we supposed that agreement or accuracy would increase as the CLP service continued in existence for longer periods, as a consequence of training and supervision provided to ward doctors on the professional skills needed to deal with medical–psychiatric comorbidity.

## 2. Materials and Methods

### 2.1. Setting and Study Design

Observations were performed at the Modena GH, a 600-bed university hospital belonging to the Regione Emilia-Romagna Health System. The GH is located in the central city area of Modena, a mid-sized city in Northern Italy (population approx. 200 thousand people). The hospital has an Emergency Department (ED), allowing direct access to patients with acute medical needs.

The Modena CLP service was instituted in 1989 and is incorporated into an operative unit of the Adult Mental Health Care Department, which also includes a 12-bed residential facility for psychiatric rehabilitation and a community mental health center. The Modena acute psychiatric ward is located in another GH of the same district, belonging to a different operative unit. The CLP service provides both routine and urgent psychiatric consultations to all wards of the GH (with the exception of the pediatric ward) and to the ED. The patients who are referred are mostly inpatients, but collaboration with a limited number of outpatient clinics (notably, oncology, liver transplantation, and dialysis) is available.

Referral forms for psychiatric consultations are sent electronically through the IT system, using a semi-structured specific form [7,26]. The service commits to see to routine referrals within 48 h, but most patients are seen within 24 h of referral. Urgent referrals may be initiated via telephone, and in this case, the service commits to see the patient within 1 h. One or two of the first assessments each day are set in advance for outpatients.

At the beginning of the working day, a staff meeting takes place to produce a group reading and distribution of the referrals received to allow tasks to be assigned. In the afternoon, team discussion of clinical cases and supervision of residents is carried out. After this, psychiatric consultation letters that specify the psychiatric diagnosis (made according to the International Statistical Classification of Diseases and Related Health Problems-10 [27]) are sent electronically, and the treatment plan is sent back to GH doctors, accompanied by verbal details over the telephone when necessary. The clinical procedures for psychiatric consultations follow international standards [28] adapted to features of the local context.

An electronic database developed in Microsoft Excel (latest version) was adopted by the CLP service to document clinical activities following the involvement of the service in research on CLP promoted in Europe by the ECLW in the late 90’s [4,5,29]. A complete, homogeneous, and reliable electronic documentation is available from year 2000. The variables in the database are organized into the following sections:(1)Socio-demographic data,(2)Details of referral,(3)Medical and psychiatric history,(4)Outcome of psychiatric assessment, and(5)Back-referral.

The same structure was followed in the description of the dataset, displayed in the results section. The data here presented refer to the period between 1 January 2000, and 31 December 2019 (or 20 full years of data).

### 2.2. Statistical Analyses

The clinical and non-clinical variables in the dataset were described using percentages for the categorical and dichotomous variables and median, mean, and SD for the continuous variables. Cohen’s Kappa statistics [30] were used to assess diagnostic concordance between diagnoses made by the CLP team and the diagnosis considered by the physician who sent the referral. The results of this statistic have been interpreted according to Cohen’s suggestion, as follow: kappa values ≤ 0.01 as indicating no agreement and 0.01–0.20 as none to slight, 0.21–0.40 as fair, 0.41–0.60 as moderate, 0.61–0.80 as substantial, and 0.81–1.00 as almost perfect agreement [30]. A difference in the kappa values reflecting a step to a higher (or lower) category of agreement was considered significant. Moreover, we deemed informative a difference of kappa values at the first decimal place, relying on the conceptual understanding of Cohen’s kappa previously proposed, assuming the squaring of the kappa as the amount of accuracy [31]. The statistical analyses were performed with Stata 15 (StataCorp, College Station, TX, USA).

### 2.3. Statement of Ethics

The study design was approved by the competent local ethical committee (Comitato Etico AVEN, Italy) on 22 September 2020 (ethical approval code: AVEN 886/2020/OSS*/AUSLMO).

## 3. Results

### 3.1. Description of the Sample

A total of 18,888 (9089 [48.1%] from 2000 to 2009 and 9799 [51.9%] from 2010 to 2019) referrals were analyzed. The majority of referrals to CLP concerned female patients, at 54% (*n* = 10,187); the median age of referred patients was 59 years (mean = 57.8; SD = 18.1), with a range of 18 to 103 years. Overall (past and/or on-going) positive psychiatric history was documented for 6940 subjects (49.4%, excluding missing values). Referrals received from medical wards were vastly more common than those received from surgical wards (13,710 vs. 3398; approximately 80% vs. 20%). The majority of referrals (*n* = 6689; 35.4%) were sent from internal (or general) medicine wards, which also accounted for 20.70% of the total beds in the hospital. The most common reason for the referrals was a request to confirm a diagnosis of depression (*n* = 4937; 32.3%), followed by requests to address agitation (*n* = 1534; 10.0%). Urgent referrals accounted for 28.0% of the total (*n* = 4622), and the average rate of admission to the psychiatric ward over the entire period of study was 2.9% (*n* = 541). A psychiatric diagnosis was established at the end of more than two-thirds of consultations (*n* = 13,883; 73.8%), but no formal psychiatric diagnosis emerged after consultation in 26.2% cases (*n* = 4828). Anxiety (*n* = 1183; 6.3%), depressive disorder (*n* = 4182; 22.1%), mixed anxiety and depression (*n* = 5365; 28.4%), and adjustment disorders (*n* = 3250; 17.2%) were by far the most common diagnoses made by CL psychiatrists, followed by delirium (*n* = 1157; 6.1%) and substance abuse disorder (*n* = 1144; 6.1%). Table 1 presents the full details of the referrals.

### 3.2. Concordance of Diagnosis between CL Psychiatrist and GH Doctor

Inter-rater agreement on the primary psychiatric diagnosis as made by the CL psychiatrist and the treating physician was evaluated with Cohen’s kappa coefficient, and the results are given in Table 2.

A diagnosis of depression was given by the referring GH doctor in 4937 cases, and in 4182, the CL psychiatrist diagnosed depression, producing a Cohen’s kappa coefficient of 0.281 for this dyad, indicating fair agreement. The diagnosis of anxiety was taken into account by the GH doctor in 1379 cases and by the CL psychiatrist in 1183, with a kappa value of 0.312. When anxiety and depression were considered together, the concordance of this broad diagnostic regrouping between GH doctor and CL psychiatrist increased up to kappa = 0.305, which still indicated fair agreement.

A diagnosis of delirium was made by the referring physician in 436 cases, and the CL psychiatrist diagnosed delirium in 1157 cases, for a kappa value of 0.342, suggesting fair agreement. Interestingly, if a broader array of symptoms associated with the diagnosis of delirium (e.g., agitation) were listed as part of the referral reason, 1970 cases were considered by the referring doctor to have delirium, which means that the level of concordance for this wider diagnosis of delirium (corresponding to the descriptive diagnosis of hyperkinetic delirium) increased to reach agreement, kappa = 0.504.

A diagnosis of alcohol or substance abuse was made in 907 cases by the GH doctor and by the CL psychiatrist in 1144; the level of concordance was kappa = 0.574, indicating moderate agreement.

Finally, a diagnosis of eating disorder was considered by GH doctor in 63 cases, and the CL psychiatrist made this diagnosis in 107 cases, when concordance was estimated, kappa value was 0.315, indicating fair agreement.

The levels of agreement in relation to different features of the referrals are presented in terms of kappa values in Table 3, Table 4, Table 5, Table 6 and Table 7.

The diagnoses of alcohol or substance abuse were generally more accurate when made by physicians than by surgeons (kappa = 0.601 and = 0.415, respectively). Conversely, surgeons seemed to be somewhat more concordant with psychiatrists than physicians were in diagnosing eating disorders (kappa = 0.545 and = 0.310, respectively) and delirium, particularly the hyperkinetic type (kappa = 0.566 and = 0.491, respectively).

For urgent and non-urgent referrals, no substantial differences were seen in the degree of diagnostic agreement that emerged for all but eating disorder diagnoses that displayed higher concordance when referred urgently (kappa = 0.432 [urgent] and 0.287 [non-urgent]).

Patients with positive psychiatric history were diagnosed with higher levels of concordance between GH doctor and CL psychiatrist for all of the disorders observed, with the exception of those that were exogenous or due to a medical condition, i.e., alcohol or substance abuse (kappa = 0.556 for both those with and without psychiatric history) and delirium (kappa = 0.560 [negative psychiatric history] and = 0.444 [positive psychiatric history]).

The concordance of diagnoses between GH doctor and the CL psychiatrist increased as time went on. Notably, a comparison of the first 10 (2000–2009) and the last 10 (2010–2019) years of activity showed higher diagnostic concordance between the referring GH doctor and CL psychiatry for anxiety, kappa = 0.175 (2000–2009) and = 0.401 (2010–2019); depression, kappa = 0.186 (2000–2009) and = 0.317 (2010–2019); eating disorders, kappa < 0.01 (2000–2009) and = 0.535 (2010–2019); delirium, kappa = 0.147 (2000–2009) and = 0.427 (2010–2019); schizophrenia, kappa = 0.119 (2000–2009) and = 0.293 (2010–2019). The agreement on alcohol or substance abuse diagnoses remained moderate, kappa = 0.593 (2000–2009) and 0.559 (2010–2019).

Finally, no substantial differences were seen in the degree of diagnostic agreement according to the gender difference of the patient for all but eating disorders, that displayed moderate agreement when diagnosed in males, whereas fair in females [kappa = 0.479 (males) and = 0.285 (females)].

## 4. Discussion

This study evaluated patterns in psychiatric morbidity at the CLP of Modena, Italy. Firstly, it assessed the reasons for referral and diagnostic concordance between the referring GH doctor (both physicians and surgeons) and the CLP team. Secondarily, it analyzed differences in diagnostic agreement according to the features of the referral (i.e., urgent or non-urgent request), of the patient (i.e., positive or negative psychiatric history), of the referring ward (i.e., internal medicine or surgical ward), and of the age of the CLP service (first 10 years or second 10 years of activity).

This study found that anxiety, depression, and adjustment disorders were by far the most common psychiatric diagnoses in patients referred to the Modena CLP service, followed by diagnoses of delirium and alcohol or other substance abuse. Lower rates were found for schizophrenia or psychosis and eating disorders. These findings are in accordance with previous reports highlighting those three diagnostic categories as the most common reasons for the psychiatric request [19]. These were among what Goldberg and Huxley called “common mental disorders” [32], as has recently been confirmed elsewhere [33,34]. The results of this study show higher diagnostic agreement for anxiety than for depression and higher agreement for mixed anxiety and depression than for depression only, suggesting that the diagnostic accuracy of affective disorders increases in the presence of anxiety. This confirms previous evidence showing difficulties in rising diagnostic concordance for depressive disorders compared with anxiety disorders [18,23]. This may be due to the clinical features of anxiety, which grow during panic attacks and commonly present with characteristic somatic symptoms that make it easier for the GH doctor to recognize the condition.

The high rates of referrals for delirium are in line with previous evidence that indicates an overall aging of the inpatient population in GHs and reveal a high incidence of delirium in the inpatient setting [35,36], a common reason for psychiatric referral [22,24,37]. The fair diagnostic accuracy found for delirium (which rose to moderate when the presentation was associated with agitation) indicates a remarkable and well-known risk that GH doctors may miss a delirium diagnosis [36,38], particularly in case of hypo-active delirium [39]. Further, the diagnostic concordance for delirium was lower in patients with a positive psychiatric history, suggesting that this fact may mislead the GH doctors and make them more likely to attribute symptoms of delirium to a previous psychiatric condition, preventing a proper recognition of the syndrome. CL psychiatrists should collaborate closely with the medical team and instruct them in the clinical skill of identifying delirium at its earliest stage to allow appropriate management to begin, which is associated with improved outcomes [40]. Further, GH multi-professional staff should be provided with more information on how to manage delirium and should be informed of the consequences of the lack of detection and management of delirium on the outcome of physical illness [41]. This may be a good incentive for them to consider the possibility of delirium, in fact it is known that one of the best model to predict higher agreement with the experts is percentage of time spent in group practice [21]. Consistent with this, the diagnostic concordance for delirium increased during the course of the CLPS’s functioning, which is evidence for effectiveness of the training and educational interventions it provided.

The findings of the present study suggest that in about a quarter of referrals (*n* = 4828, 26.2%) no psychiatric diagnosis was formalized by CL psychiatrists. In most of these cases, the referring doctor relied on general or descriptive vocabulary that could denote a mental disorder or described the need of the GH doctor (i.e., “psychiatric history,” “non-compliant,” or “refused medical treatment”). Frequently, the GH doctors were requesting “psychiatric clearance” before major surgery (e.g., pre-orthotopic liver transplantation or tocophobia) or before discharging vulnerable patients, that is, to rule out significant psychiatric problems in general, or simply on the basis that the patient was already known to be suffering from a mental disorder. These findings, already pinpointed by Clarke et al. as the “staff problems” reason for referral [24], could be interpreted in multiple ways. First, they indicated an unmet need for psychiatric diagnostic skills in referring doctors [42]. Second, these types of referral, which led to diagnostic disagreement, may reflect the fact that the GH doctor did not feel at ease or possibly felt challenged in dealing with medical–psychiatric comorbidity, or, again, that they were worried about possible medico-legal implications (e.g., patients refusing medical treatments, or requesting early discharge from GH), thus requesting a psychiatric consultation to share the risk for difficult clinical management decisions [8]. Accordingly, the findings of the present study reflect the reality of routine clinical practice and provide guidance for the role of CL psychiatrist, who can bridge gaps in knowledge and skills, empowering physicians and surgeons to deal with challenging clinical situation. Indeed, a large portion of the *liaison* activity of a CLP service should include day-to-day professional support and training for GH doctors to reduce stigma against the mental health problems of their patients, which can lead to worse clinical outcomes if neglected [43,44,45]. While this is only an indicator, the present study provides hints of improvement in this direction, in the form of an increased diagnostic agreement between GH doctors and CL psychiatrist over the second period of the operation of CLP service than during the first half, namely, after 10 years of liaison activity, indicating an increase in trust and reciprocity between professionals in the care of the hospitalized people.

For specific diagnoses, it has been found that the GH doctors were more accurate when they suspected an affective disorder that was comorbid with anxiety and were more accurate for delirium when it was hyper-kinetic. This has at least two possible explanations. First, GH doctors are better able to recognize psychiatric issues when behavioral symptoms occur (such as agitation or panic) and fail to properly assess negative symptoms (such as lack of volition or depressed mood). Second, the results may reflect intrinsic limits of the categorical approach to diagnoses of mental disorder: a growing body of literature suggests the need to shift to a more dimensional or functional understanding of mental disorders [46,47]. Here, it is worth mentioning that the Research Domain Criteria (RDoC) framework can be used in both clinical and research practice to increase diagnostic accuracy and reliability [48,49].

Finally, diagnostic concordance was greater for substance or alcohol abuse and for eating disorders, suggesting that GH doctors are better equipped to recognize psychiatric conditions in presence of exogenous etiology and/or clear clinical or radiological signs, such as liver injury for alcoholism or impaired BMI for eating disorders. The greater diagnostic concordance highlighted for substance and alcohol abuse is consistent with the most recent literature [18].

Notably, the gender difference in the diagnostic agreement found for eating disorders, favoring males, is surprising since a body of literature suggested a higher prevalence of these conditions among females than males, and this finding may be influenced by the low number of diagnoses recorded in this category [50,51]. Notwithstanding, recent studies have been suggesting that at the community level, a higher percentage of males than females with eating disorders met criteria for an urgent medical inpatient admission [52]. Our results may be interpreted also consistent with this, suggesting either delayed treatment seeking, a more rapid escalation into medical instability, or both, among males with eating disorders, leading them to be more likely to be seen by CL psychiatrist within GH. Conversely, no agreement was found for somatization or medically unexplained symptoms, confirming the presence of well-known pitfalls to proper identification.

### Limitations

Certain limitations of this study must be acknowledged. First, although the data were gathered from a large number of consultations, the study is monocentric, so its results may not be generalizable to other centers, particularly those without a dedicated CLP service. Second, the retrospective design of this study may have affected the accuracy of data collection, although the dataset was double-checked by two researchers (M.M. and L.P.) before the analyses were performed and missing or uncertain data were removed. Third, in such a wide timespan (20 years) the inter-rater agreement may have changed also as an effect of the succeeding of different CL psychiatrists. Nevertheless, the Modena CLP service has always been characterized by a shared bio-psycho-social framework and a psychodynamic orientation that have been source of consistency and diagnostic stability across the years, for similar clinical conditions [53]; also, one of the authors (S.F.) has been part of the staff of the CLP service for the whole 20 years, contributing further to stabilize diagnostic procedures and methods of data collection. Fourth, the psychiatric referrals were requested using a semi-structured form, which provided a defined clinical question and a diagnostic hypothesis. However, the referring doctors were not informed that the accuracy of their diagnostic orientation would be assessed. Even if this blinding of the referring GH doctors may have increased the truthfulness of the results, it is plausible that higher diagnostic concordance would have been found if ward clinicians had been informed that their referrals would be judged for accuracy, which might have encouraged them to be as accurate as possible as they formalized their diagnostic hypothesis. Finally, the study did not evaluate the impact of psychiatric morbidity on clinical outcome of the patient. Future studies should attempt to overcome these limitations.

## 5. Conclusions

Diagnostic agreement about the psychiatric diagnoses of medically ill patients staying in the GH between ward doctors (both clinicians and surgeons) and CL psychiatrists ranged from fair to moderate. Referring doctors more accurately recognized a psychiatric condition when positive symptoms (such as panic or agitation) occurred or when they relied on exogenous etiology or on clear clinical signs, while negative symptoms (such as lack of volition or depressed mood) or complex psychiatric conditions (such as psychoses or MUS) were more likely to be misdiagnosed. The results of this study indicate that close day-to-day collaboration between CL psychiatrists and GH doctors produced improvements in their ability to detect comorbid psychiatric conditions. Future studies should replicate these findings at other centers, challenging GH doctors to be more actively engaged in the psychiatric comorbidity recognition and evaluating the impact of psychiatric morbidity and its recognition on clinical outcomes. Furthermore, future studies should examine the implementation of a more dimensional and functional diagnostic approach at GHs, such as the RDoC framework, to increase diagnostic accuracy. Occasions for shared training between GH doctors and CL psychiatrists should be pursued to increase respective skills and collaboration on specific cases, leading to better care integration.

## Figures and Tables

**Table 1 ijerph-18-00749-t001:** Details of referrals.

Variable	*N*	%
Gender		
Female	10187	54.0
Male	8685	46.0
Psychiatric anamnesis		
Negative	7097	50.6
Positive	6940	49.4
Urgency		
Non-urgent	11891	72.0
Urgent	4622	28.0
Referring ward		
Medical wards	13710	80.0
Surgical wards	3398	20.0
Post-referral psychiatric admission		
No	18347	97.1
Yes	541	2.9
Reason for referral		
Depression	4937	26.1
Agitation	1534	8.1
Anxiety	1379	7.3
Medically unexplained symptoms (MUS)	1240	6.6
Alcohol or substance abuse	907	4.8
Delirium	436	2.3
Psychosis	173	0.9
Dementia	78	0.4
Eating disorder	63	0.3
Other	8141	43.1
CL psychiatrist diagnosis		
Depression	4182	22.1
Adjustment disorder	3250	17.2
Anxiety	1183	6.3
Delirium	1157	6.1
Alcohol or substance abuse	1144	6.1
Dementia	957	5.1
Schizophrenia or psychosis	531	2.8
Somatization	218	1.2
Eating disorder	107	0.6
Other	1154	6.1
None	4828	26.2
	Mean	SD (Range)
Age	57.8	18.1 (18–103)

*N*: Number (Frequency); %: percentages have been calculated on the subtotals, excluding missing values; CL: Consultation-liaison; SD: Standard deviation.

**Table 2 ijerph-18-00749-t002:** Overall agreement on primary psychiatric diagnoses by CL psychiatrists and treating physicians.

Reason for Referral	Diagnosis	Cohen’s Kappa
Abuse or intoxication	Alcohol or substance abuse/dependence	0.574
Anxiety	- Anxiety	0.312
	- Depression	0.010
	- Anxiety and depression	0.134
	- Adjustment disorder	0.018
Depression	- Anxiety	<0.01
	- Depression	0.281
	- Anxiety and depression	0.223
	- Adjustment disorder	0.225
Anxiety and depression	- Anxiety	0.069
	- Depression	0.260
	- Anxiety and depression	0.305
	- Adjustment disorder	0.207
Eating disorder	Eating disorder	0.315
- Agitation	Delirium	0.371
- Delirium		0.342
- Agitation and delirium		0.504
- Agitation	Dementia	0.156
- Delirium		0.030
- Dementia		0.115
Psychosis	Schizophrenia or psychosis	0.202
MUS	Somatization	0.159

MUS: Medically unexplained symptoms.

**Table 3 ijerph-18-00749-t003:** Comparison of inter-rater agreement: Referrals from medical wards vs. surgical wards.

Reason for Referral	Diagnosis	Cohen’s Kappa Medical	Cohen’s Kappa Surgical
Abuse or intoxication	Alcohol or substance abuse or dependence	0.601	0.415
Anxiety	- Anxiety	0.322	0.319
	- Depression	<0.01	<0.01
	- Anxiety and depression	0.062	0.113
	- Adjustment disorder	0.019	<0.01
Depression	- Anxiety	0.008	<0.01
	- Depression	0.284	0.258
	- Anxiety and depression	0.263	0.224
	- Adjustment disorder	0.225	0.289
Anxiety and Depression	- Anxiety	0.135	0.153
	- Depression	0.224	0.213
	- Anxiety and depression	0.308	0.295
	Adjustment disorder	0.207	0.267
Eating disorder	Eating disorder	0.310	0.545
- Agitation	Delirium	0.364	0.414
- Delirium		0.353	0.345
- Agitation and delirium		0.491	0.566
- Agitation	Dementia	0.177	<0.01
- Delirium		<0.01	<0.01
- Dementia		0.136	<0.01
Psychosis	Schizophrenia or psychosis	0.244	0.132
MUS	Somatization	0.175	0.084

MUS: Medically unexplained symptoms.

**Table 4 ijerph-18-00749-t004:** Comparison of inter-rater agreements: Urgent vs. non-urgent requests.

Reason for Referral	Diagnosis	Cohen’s Kappa Urgent	Cohen’s Kappa Non-Urgent
Abuse or intoxication	Alcohol or substance abuse or dependence	0.542	0.583
Anxiety	- Anxiety	0.386	0.274
	- Depression	<0.01	<0.01
	- Anxiety and depression	0.188	<0.01
	- Adjustment disorder	<0.01	<0.01
Depression	- Anxiety	<0.01	<0.01
	- Depression	0.278	0.252
	- Anxiety and depression	0.267	0.221
	- Adjustment disorder	0.166	0.206
Anxiety and depression	- Anxiety	0.221	<0.01
	- Depression	0.206	0.200
	- Anxiety and depression	0.371	0.250
	- Adjustment disorder	0.133	0.198
Eating disorder	Eating disorder	0.432	0.287
- Agitation	Delirium	0.382	0.342
- Delirium		0.286	0.379
- Agitation and delirium		0.498	0.497
- Agitation	Dementia	0.128	0.158
- Delirium		<0.01	<0.01
- Dementia		<0.01	0.151
Psychosis	Schizophrenia or psychosis	0.216	0.215
MUS	Somatization	0.185	0.151

MUS: Medically unexplained symptoms.

**Table 5 ijerph-18-00749-t005:** Comparison of inter-rater agreement: Positive psychiatric history vs. negative psychiatric history.

Reason for Referral	Diagnosis	Cohen’s Kappa Psy History+	Cohen’s KappaPsy History−
Abuse or intoxication	Alcohol or substance abuse or dependence	0.556	0.556
Anxiety	- Anxiety	0.378	0.264
	- Depression	<0.01	<0.01
	- Anxiety and depression	<0.01	<0.01
	- Adjustment disorder	<0.01	<0.01
Depression	- Anxiety	<0.01	<0.01
	- Depression	0.312	0.194
	- Anxiety and depression	0.283	0.166
	- Adjustment disorder	0.195	0.224
Anxiety and depression	- Anxiety	0.123	0.113
	- Depression	0.255	0.122
	- Anxiety and depression	0.348	0.192
	- Adjustment disorder	0.183	0.212
Eating disorder	Eating disorder	0.524	0.125
- Agitation	Delirium	0.346	0.434
- Delirium		0.310	0.292
- Agitation and delirium		0.444	0.560
- Agitation	Dementia	0.179	<0.01
- Delirium		<0.01	<0.01
- Dementia		0.142	0.134
Psychosis	Schizophrenia or psychosis	0.229	0.191
MUS	Somatization	0.240	0.110

MUS: Medically unexplained symptoms.

**Table 6 ijerph-18-00749-t006:** Comparison of inter-rater agreement: 2000–2009 (first 10 years) vs. 2010–2019 (second 10 years).

Reason for Referral	Diagnosis	Cohen’s Kappa 2000–2009	Cohen’s Kappa 2010–2019
Abuse or intoxication	Alcohol or substance abuse or dependence	0.593	0.559
Anxiety	- Anxiety	0.175	0.401
	- Depression	<0.01	<0.01
	- Anxiety and depression	<0.01	<0.01
	- Adjustment disorder	<0.01	<0.01
Depression	- Anxiety	<0.01	<0.01
	- Depression	0.186	0.317
	- Anxiety and depression	0.180	0.282
	- Adjustment disorder	0.198	0.244
Anxiety and depression	- Anxiety	<0.01	0.160
	- Depression	0.148	0.249
	- Anxiety and depression	0.199	0.351
	- Adjustment disorder	0.188	0.222
Eating disorder	Eating disorder	<0.01	0.535
- Agitation	Delirium	0.147	0.427
- Delirium		0.335	0.343
- Agitation and delirium		0.334	0.554
- Agitation	Dementia	0.185	0.141
- Delirium		<0.01	<0.01
- Dementia		<0.01	0.189
Psychosis	Schizophrenia or psychosis	0.119	0.293
MUS	Somatization	0.141	0.227

MUS: Medically unexplained symptoms.

**Table 7 ijerph-18-00749-t007:** Comparison of inter-rater agreement: Male vs. female patients.

Reason for Referral	Diagnosis	Cohen’s Kappa Male	Cohen’s Kappa Female
Abuse or intoxication	Alcohol or substance abuse or dependence	0.588	0.537
Anxiety	- Anxiety	0.313	0.309
	- Depression	<0.01	<0.01
	- Anxiety and depression	0.071	0.065
	- Adjustment disorder	0.050	<0.01
Depression	- Anxiety	0.022	<0.01
	- Depression	0.287	0.268
	- Anxiety and depression	0.267	0.243
	- Adjustment disorder	0.282	0.180
Anxiety and depression	- Anxiety	0.152	0.118
	- Depression	0.232	0.206
	- Anxiety and depression	0.310	0.288
	- Adjustment disorder	0.279	0.153
Eating disorder	Eating disorder	0.479	0.285
- Agitation	Delirium	0.394	0.328
- Delirium		0.312	0.380
- Agitation and delirium		0.515	0.483
- Agitation	Dementia	0.132	0.185
- Delirium		0.022	0.037
- Dementia		0.102	0.127
Psychosis	Schizophrenia or psychosis	0.241	0.163
MUS	Somatization	0.175	0.150

MUS: Medically unexplained symptoms.

## Data Availability

Data sharing with investigators outside the team requires Ethical Committee approval. Data requests may be submitted to the corresponding author. All request for anonymized data will be reviewed by the research team and then submitted to the AVEN Ethical Committee.

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
