# Peer review of "Diagnostic Agreement between Physicians and a Consultation–Liaison Psychiatry Team at a General Hospital: An Exploratory Study across 20 Years of Referrals"

_ijerph, 2021, doi:10.3390/ijerph18020749_

Round 1
Reviewer 1 Report
A well conducted study with a large sample on a relevant clinical topic. Remarkably well written. In line 174: the word "made" probably needs to be deleted.
Author Response
A well conducted study with a large sample on a relevant clinical topic. Remarkably well written. In line 174: the word "made" probably needs to be deleted.
Corrected as suggested. A few more typos and mistakes were corrected, these were located at: page 2, lines 95-100, continuing to page 3 lines 122-123; page 4 line 186; page 9 lines 279-281.
Finally, at page 11, lines 401-430 the following comment has been added to improve the meaning of the sentence: “[…] also, one of the authors (S.F.) has been part of the staff of the CLP service for the whole 20 years, contributing further to stabilize diagnostic procedures and methods of data collection.”
We thank the anonymous Reviewer for her/his suggestion
Reviewer 2 Report
This study addresses the activities of the CLP Service over a period of 20 years and the rate of concordance of diagnosis between physicians and the CLP team. Although it is limited to single center results, it confirms data from over 18,000 patients, a number that overwhelms the previous literature. Basically, I would like to see this paper published in International Journal of Environmental Research and Public Health, but still, I have serious concerns on several issues in this paper.
A critical Issue.
A major problem with this study is that there is little mention in either the introduction or discussion of previous literature that investigate the rate of diagnostic agreement between physicians and CLPs.
In line 79, it is stated that "On the basis of our experience, we hypothesized that affective disorders would be the most frequent reason for referral and that these referrals would have low diagnostic accuracy", however, it is not appropriate to base this on the authors' experience. Sufficient mention should be made in the introduction of the fact that several previous publications have already discussed the diagnostic concordance rate.
The following are some of the studies that I found in my hand search. There may be others that I just can't find.
Studies focusing on depression and anxiety:
https://doi.org/10.1080/13607863.2015.1063103
https://doi.org/10.1176/appi.psy.44.5.407
https://doi.org/10.1016/0022-3999(94)00127-Q
https://doi.org/10.1016/S0033-3182(96)71549-9
Studies examining the overall agreement rate:
https://doi.org/10.1177/0091217420905462
https://doi.org/10.1097/NMD.0000000000001018
https://doi.org/10.1111/j.1447-0594.2011.00771.x
https://doi.org/10.1017/S147895151700027X
https://doi.org/10.1186/s13030-020-00188-6
https://doi.org/10.3109/00048679509064950
https://doi.org/10.1111/j.1440-1819.2011.02272.x
Furthermore, in the Discussion, the differences between the results of these previous literatures and the results of this study should also be comprehensively mentioned.
This study is superior to those in the previous literature in that the number of subjects is much larger, so I still believe this study is of high value.
Minor issues.
- There are many similar abbreviations such as CL, CLP, and CLPS, which I think confuse readers. At least for CLPS, it would be easier to read if the authors write “CLP service”.
- I think the phrase "CLP psychiatrist" in line 74 is incorrect.
Author Response
Reviewer 2
In line 79, it is stated that "On the basis of our experience, we hypothesized that affective disorders would be the most frequent reason for referral and that these referrals would have low diagnostic accuracy", however, it is not appropriate to base this on the authors' experience. Sufficient mention should be made in the introduction of the fact that several previous publications have already discussed the diagnostic concordance rate.
The following are some of the studies that I found in my hand search. There may be others that I just can't find...
(refs)
Furthermore, in the Discussion, the differences between the results of these previous literatures and the results of this study should also be comprehensively mentioned.
We wish to thank very much the Reviewer for her/his accurate suggestion, that we have followed by including reference and comment to the mentioned papers in the introduction and discussion sections of the original manuscript.
- References have been added as: https://doi.org/10.1177/0091217420905462 [18]; https://doi.org/10.1097/NMD.0000000000001018 [19]; https://doi.org/10.1080/13607863.2015.1063103 [20]; https://doi.org/10.1016/S0033-3182(96)71549-9 [21]; https://doi.org/10.1111/j.1440-1819.2011.02272.x [22]; https://doi.org/10.1016/0022-3999(94)00127-Q [23]; https://doi.org/10.3109/00048679509064950 [24]; https://doi.org/10.1176/appi.psy.44.5.407 [25]; https://doi.org/10.1111/j.1447-0594.2011.00771.x [37]; https://doi.org/10.1186/s13030-020-00188-6 [38]; https://doi.org/10.1017/S147895151700027X [39].
- Page 2, line 73-75, we have added the following sentence: “These few showed variable agreement and disagreement rates dependent on multiple factors, such as the diagnostic category [18,19], the age of the patient [20], and the professional experience of the physician [21]”;
- Page 2, line 87, we have added the following comment: “Taking into consideration the available literature, […]”, and at line 89 the references to studies [18-10, 22-25];
- Page 10, lines 294-295, we have added the following comment: “These findings are in accordance with previous reports highlighting those three diagnostic categories as the most common reasons for the psychiatric request [19].”;
- Page 10, lines 300-301, we have added the following sentence: “This confirms previous evidence showing difficulties in rising diagnostic concordance for depressive disorders compared with anxiety disorders [18,23]”;
- Page 10, lines 304-309, we have improved the following sentences: “The high rates of referrals for delirium are in line with previous evidence that indicates an overall aging of the inpatient population in GHs and reveal a high incidence of delirium in the inpatient setting [35,36], a common reason for psychiatric referral [22,24,37]. The fair diagnostic accuracy found for delirium (which rose to moderate when the presentation was associated with agitation) indicates a remarkable and well-known risk that GH doctors may miss a delirium diagnosis [36,38], particularly in case of hypo-active delirium [39].”;
- Page 10, line 312, we have added the word “previous”;
- Page 10, lines 318-320, we have added the following comment: “[…] in fact it is known that one of the best model to predict higher agreement with the experts is percentage of time spent in group practice [21].”
- Page 10, lines 330-331, we have added the following comment: “These findings, already pinpointed by Clarke et al. as the “staff problems” reason for referral [24], […]”;
- Page 11, line 378-379, we have added the following sentence: “The greater diagnostic concordance highlighted for substance and alcohol abuse is consistent with the most recent literature [18].”
Minor issues.
1. There are many similar abbreviations such as CL, CLP, and CLPS, which I think confuse readers. At least for CLPS, it would be easier to read if the authors write “CLP service”.
Acronyms and abbreviations have been revised and simplified, notably CLPS has been replaced with CLP service all across the manuscript, as suggested by the Reviewer.
2. I think the phrase "CLP psychiatrist" in line 74 is incorrect
The Reviewer is absolutely right, this was corrected.
Reviewer 3 Report
This study used 20 years of referral data from a hospital to examine the concordance between physicians and the consultation–liaison psychiatry team. The studies found fair to moderate concordance in several common types of psychiatric conditions and some disparities by time, physician specialties, and types of psychiatric disorders. The authors did a good job presenting the results. The discussion part was very well done as it helps the readers better make sense of the findings and discussed implications for future research and clinical practice.
A few minor issues:
- Did the authors examine any differences by patients' gender? Studies have suggested gender differences in the symptomatology of depression and other types of psychiatric disorders. Might be important to address potential gender differences.
- The comparisons of cohen's kappa, especially the interpretation of the magnitude of differences between two statistics, seemed arbitrary. This is to no fault of the authors, in a sense. However, the authors might consider mentioning a threshold or standard used to talk about whether one kappa is much higher (or lower) than the other one. 10% difference? 20%?
Author Response
Reviewer 3
1. Did the authors examine any differences by patients' gender? Studies have suggested gender differences in the symptomatology of depression and other types of psychiatric disorders. Might be important to address potential gender differences.
We thank the Reviewer for this input. We have added the comparison of the diagnostic agreement according to the gender of the patient.
- Page 2, line 84 among the features of the patient considered we specified: “gender, and”;
- Page 6, line 234: we changed “Table 3-6” to “Table 3-7”;
- Table 7, reporting the Comparison of inter-rater agreement between male and female patients, has been added at page 8, line 249;
- Page 9, line 266: the word “Finally” has been deleted;
- Page 9, lines 274-277: the following statement has been added “Finally, no substantial differences were seen in the degree of diagnostic agreement according to the gender difference of the patient for all but eating disorders that displayed moderate agreement when diagnosed in males, whereas fair in females (kappa = 0.479 [males] and = 0.285 [female]).”;
- Page 11, lines 380-388: we have added the discussion of the results of the comparison of the agreement according to the gender
“Notably, the gender difference in the diagnostic agreement found for eating disorders, favoring males, is surprising since a body of literature suggested a higher prevalence of these conditions among females than males, and this finding may be influenced by the low number of diagnoses recorded in this category [50, 51]. Notwithstanding, recent studies have been suggesting that at the community level, a higher percentage of males than females with eating disorders met criteria for an urgent medical inpatient admission [52]. Our results may be interpreted also consistent with this, suggesting either delayed treatment seeking, a more rapid escalation into medical instability, or both, among males with eating disorders, leading them to be more likely to be seen by CL psychiatrist within GH.”
Relevant references 50-52 have been added.
2. The comparisons of cohen's kappa, especially the interpretation of the magnitude of differences between two statistics, seemed arbitrary. This is to no fault of the authors, in a sense. However, the authors might consider mentioning a threshold or standard used to talk about whether one kappa is much higher (or lower) than the other one. 10% difference? 20%?
We thank the Reviewer for the opportunity to clarify this methodological aspect. In the “2.2 Statistical analyses” subsection of “2. Materials and Methods” we have added the explanation of the interpretation of the Cohen’s kappa statistic used, and added one relevant reference [31]. Please, find at page 3, lines 159-166: “The results of this statistic have been interpreted according to Cohen’s suggestion, as follow: kappa values ≤ 0.01 as indicating no agreement and 0.01–0.20 as none to slight, 0.21–0.40 as fair, 0.41– 0.60 as moderate, 0.61–0.80 as substantial, and 0.81–1.00 as almost perfect agreement [30]. A difference in the kappa values reflecting a step to a higher (or lower) category of agreement was considered significant. Moreover, we deemed informative a difference of kappa values at the first decimal place, relying on the conceptual understanding of Cohen's kappa previously proposed, assuming the squaring of the kappa as the amount of accuracy [31].”
Round 2
Reviewer 2 Report
The manuscript has been much improved and is in a nice condition now.